# Effects of age on non-communicable disease risk factors among Nepalese adults

**Bhim Prasad Sapkota**[1,2]*, **Kedar Prasad Baral**[3], **Eva A. Rehfuess**[4,5], **Klaus G. Parhofer**[6], **Ursula Berger**[4,5]

**1** CIH<sup>LMU</sup> Center for International Health, LMU Munich, Munich, Germany, **2** Teaching & Training Unit, Division of Infectious Diseases and Tropical Medicine, University Hospital, LMU Munich, Munich, Germany, **3** Patan Academy of Health Sciences, Lalitpur, Nepal, **4** Institute of Medical Information Processing, Biometry and Epidemiology, LMU Munich, Munich, Germany, **5** Pettenkofer School of Public Health, Munich, Germany, **6** Medical Department-4, University Hospital, LMU Munich, Munich, Germany

* bhim.sapkota@lrz.uni-muenchen.de, bhimprasadsapkota@gmail.com

## Abstract

The growing burden of non-communicable diseases (NCDs) and an increase in the prevalence of the underlying risk factors are creating a challenge to health systems in low- and middle-income countries (LMICs). In Nepal, deaths attributable to NCDs have been increasing, as has life expectancy. This poses questions with regards to how age and various risk factors interact in affecting NCDs. We analyzed the effects of age on NCD risk factors, using data from the Nepalese STEPs survey 2019, a nationally representative cross-sectional study. Six sociodemographic determinants, four behavioral risk factors, and four biological risk factors were examined. Age effects were analyzed among three age groups: below 35 years (young), 35–59 years (middle aged) and 60 years and above (elderly). The prevalence of selected behavioral risk factors for NCDs, notably smoking, alcohol consumption and insufficient physical activity, and some biological risk factors (hypertension, hyperlipidemia) increases with age. The prevalence of most behavioral risk factors was highest among men and women aged 60 years and above. The prevalence of hypertension and hyperlipidemia was highest among the elderly, but the prevalence of diabetes and overweight/obesity was highest among the middle aged for both sexes. Age interactions in the association between behaviors and biological risk factors were surprisingly weak. However, age interactions were significant in the association between alcohol consumption and -hypertension, -overweight/obesity and -hyperlipidemia among women. While the prevalence of NCD risk factors tends to be higher among elders, the interaction between age and risk factors is complex. Most NCD risk factors are related to behaviors, which originate in young adulthood. It is necessary to diagnose and treat biological risk factors, in younger age groups before they manifest as NCDs. Similarly, behavior change interventions need to target these younger age groups to reduce the risk of NCDs later in life.

**Data Availability Statement:** All relevant data are within the paper and its Supporting Information files.

**Funding:** The author(s) received no specific funding for this work.

**Competing interests:** The authors have declared that no competing interests exist.

## Background

There is a growing pandemic of non-communicable diseases (NCDs), such that cardiovascular diseases, diabetes, chronic respiratory diseases and cancers are now the leading causes of global deaths [1]. The global NCD burden is high and the NCD share of global deaths increased from 68% in 2012 to 71% in 2016 [2] and 74% in 2019 [3]. However, NCDs are not inevitable [4]. It is estimated that 17 million NCD-related deaths occur before the age of 70 and are largely preventable [5]. Most NCDs are initiated in early life and occur as the result of an unhealthy lifestyle [6]. Modifiable behavioral risk factors (i.e. tobacco use, harmful use of alcohol, unhealthy diet and physical inactivity) and metabolic biological risk factors (i.e. raised blood pressure and blood glucose, abnormal lipids and overweight or obesity) contribute the most to the development of NCDs [7]. The prevalence of these risk factors is boosted by an aging population, an increasing life expectancy, and by urbanization [8]. Aging and an unhealthy lifestyle are the major driving factors for common NCDs in modern-day society [9]. Moreover, there is a socio-economic gradient in the presence of these risk factors within any given population [10].

The burden of NCDs is especially high in low- and middle-income countries (LMICs), where the majority (86%) of premature NCD-related deaths globally occur [5]. In low-income countries, 22% of men and 35% of women are losing their life due to NCD-attributable premature deaths, while in high-income countries the corresponding numbers are 8% and 10%, respectively [4]. In Nepal, risk factors beyond the above described classical NCD risk factors, such as household air pollution from solid fuel use, poor oral health and high salt consumption, further contribute to the development of NCDs [11, 12]. Estimated premature deaths attributed to NCDs in Nepal increased from 51% in 2010, to 60% in 2014 and further to 66% in 2016 [13]. Life expectancy for Nepalese people has increased from 67 years in 2011 [14] to 71 years in 2019 [15]. Aging is inevitable but the higher prevalence of NCDs in an ageing population creates a challenge for health systems [16].

Although demographic aging has been recognized as a key risk factor for the development of NCDs in LMICs [17], it has not been explored how age affects and interacts with NCD risk factors in the Nepalese context. As a member state of the World Health Organization (WHO), Nepal is committed to achieving the WHO's 25 by 25 strategic health goals for a reduction of NCD-related premature deaths by 25% by 2025 [18], but effective public health interventions targeting the aging population are yet to be designed and executed. Similarly, the UN Sustainable Development Goal (SDG) target 3.4 [19] calls for all member states to reduce premature mortality from NCDs by a third between 2015 and 2030. Recent reports noted, however, that most LMICs are off track to reach SDG target 3.4 for NCD mortality [20].

This study aims to assess the effects of age on behavioral and biological risk factors of NCDs by exploring the following research questions:

- How do behavioral and biological risk factors vary with age?

- How does age modify the effect of health behavior on biological risk factors?

## Methods

### Study design and study population

This cross-sectional study is based on data from the STEPs survey 2019, a nationally representative population-based household survey of adults aged 15–69 years targeting NCD risk factors and conducted by the Nepal Health Research Council [21]. Using a multistage cluster sampling technique, from each of the seven Nepalese provinces, 37 community clusters were randomly selected, thus resulting in 259 primary sampling units (PSU). From each PSU, 25

randomly selected households were included in the study and one adult per household was enrolled.

Data collection was interview-based, used the standardized NCD STEPS questionnaire version 3.2 [22] and was conducted between February and May 2019 in three phases. The first and second phases were performed at the participant's home, where socio-demographic characteristics and self-reported behavioral data were collected and physical measurements for height, weight and blood pressure were taken. The third phase was performed at a health facility located within the community cluster [21]. The overall response rate to the STEPs survey was 86.4% [21].

## Study variables

Our study includes three categories of variables based on an underlying conceptual framework as illustrated in Fig 1.

Details about the definition and levels of these binary and categorical variables are described in the (S1 File).

## Data analysis

Statistical analyses were conducted using the statistical software R, version R 4.1.1 [23] using the libraries survey [24] and mgcv [25]. Analysis used inverse probability weighting based on the sampling probabilities to adjust for differences in the age and sex composition of the sample population compared to the Nepalese population [26]. There was minimal missing data, therefore the analyses are based on complete cases. All analyses were stratified by sex.

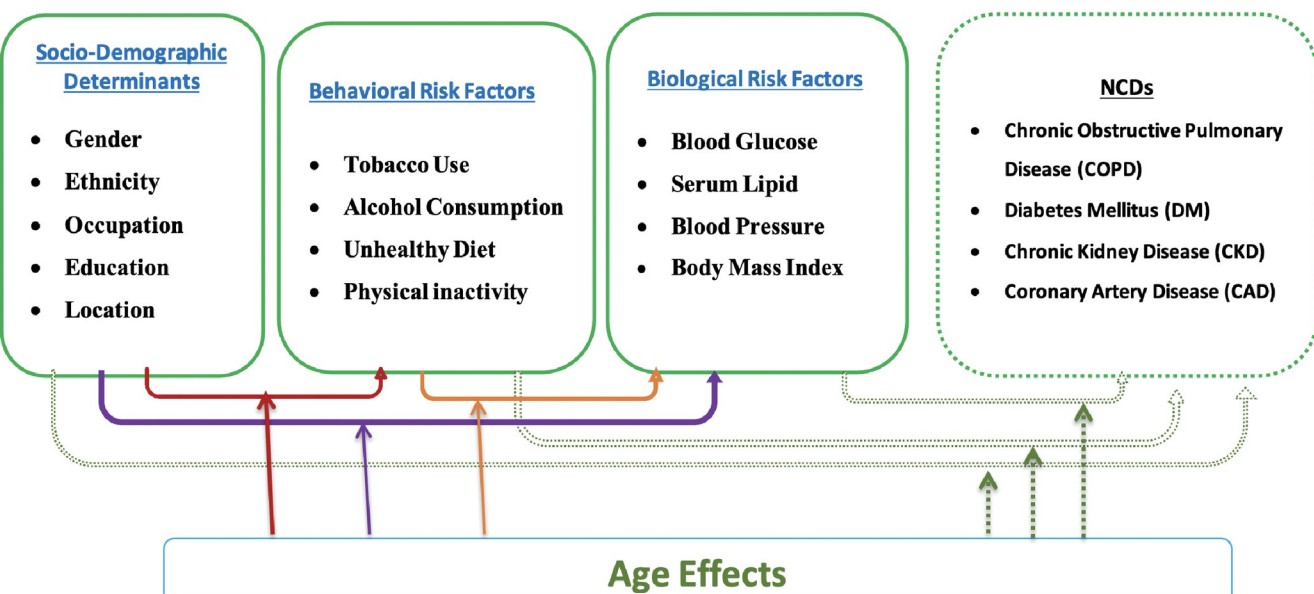

**Fig 1. Conceptual framework of the study.** *Sociodemographic characteristics*: age, sex, ethnicity, highest level of education, current occupation, wealth index and urban or rural location of the community. *Behavioral risk factors*: daily smoking, any alcohol consumption in the last 12 months, insufficient fruit and vegetable consumption, insufficient physical activity. *Biological risk factors*: overweight or obesity measured by BMI, hypertension (raised blood pressure or current use of antihypertensive drugs), hyperglycemia (raised blood glucose or current use of hypoglycemic drugs) and hyperlipidemia (raised total cholesterol or current use of lipolytic drugs).

**Table 1. Sociodemographic characteristics of participants—stratified by age group (unweighted).**

| Demographic variables | aggregate frequencies (unweighted percentage) (n = 5593) | stratified frequencies (unweighted percentage) | | |
|---|---|---|---|---|
| | | Below 35 years (n = 2196) | 35–59 years (n = 2705) | 60 years and above (n = 692) |
| **Gender** | | | | |
| Men | 1998(35.7) | 647(29.4) | 1040(38.4) | 311(44.9) |
| Women | 3595(64.3) | 1549(70.6) | 1665(61.6) | 381(55.1) |
| **Ethnic Group** | | | | |
| Dalit | 766(13.7) | 347(15.8) | 326(12.0) | 93(13.4) |
| Disadvantaged Janjati | 1640(29.3) | 634(28.8) | 809(29.9) | 197(28.4) |
| Religious Minorities | 174(3.1) | 83(3.7) | 76(2.8) | 15(2.1) |
| Advantaged Janajati | 899(16.1) | 327(14.8) | 458(16.9) | 114(16.4) |
| Upper Caste | 2124(37.8) | 805(36.6) | 1036(38.2) | 273(39.4) |
| **Level of education** | | | | |
| Primary incomplete | 2792(49.9) | 580(26.4) | 1632(60.3) | 580(83.9) |
| Primary complete | 1051(18.8) | 520(23.6) | 463(17.1) | 68(9.8) |
| Secondary | 1573(28.1) | 990(45.0) | 547(20.22) | 36(5.2) |
| University | 176(3.1) | 106(4.8) | 63(2.3) | 7(1.01) |
| **Location** | | | | |
| Rural | 2133(38.1) | 824(37.5) | 1052(38.8) | 257(37.1) |
| Urban | 3460(61.9) | 1372(62.5) | 1653(61.2) | 435(62.9) |
| **Strata** | | | | |
| Rural Municipality | 2133(38.1) | 825(37.5) | 1052(38.8) | 257(37.1) |
| Sub-/Metropolitan City | 705(12.6) | 282(12.8) | 346(12.8) | 77(11.1) |
| Municipality | 2755(49.3) | 1090(49.7) | 1307(48.4) | 358(51.7) |
| **Wealth Index** | | | | |
| Lower Wealth Index | 2715(48.5) | 1035(47.2) | 1320(48.7) | 360(52.0) |
| Middle Wealth Index | 949(16.9) | 384(17.4) | 451(16.6) | 114(16.5) |
| Upper Wealth Index | 1929(34.6) | 777(35.4) | 934(34.7) | 218(31.5) |

Distributions of sociodemographic characteristics of the study sample are reported unweighted in Table 1. The prevalence of behavioral and biological risk factors was estimated by weighted frequencies and these are reported together with their Standard Errors (SE) in Table 2. To investigate differences in the magnitude and pattern of the NCD risk factors by age, descriptive and prevalence estimates are stratified by the age groups "15–35 years", "35–59 years" and "60 years and above", and p-values based on Chi-square tests are reported.

Weighted multilevel logistic regression for clustered survey data was used to assess the association of sociodemographic characteristics with behavioral risk factors (Table 3) and the association of behavioral risk factors with biological risk factors (Table 4). Age was included in all models as a categorical variable to investigate age effects. Regression models were fitted without and with adjustment for sociodemographic variables. Odds Ratios (OR) and adjusted Odds Ratios (AOR) are reported together with 95% confidence intervals.

## Ethical approval

Permission to use the 'STEPS survey-2019 dataset' for this study was obtained from the Nepal Health Research Council (NHRC) by letter (reference no. 786). Ethical approval for the study was obtained from the NHRC (reference no. 2882) and from the ethical committee of the

**Table 2. Prevalence of behavioral and biological risk factors—stratified by sex and age group (weighted).**

| Behavioral and Biological variables | Frequencies (Weighted Percentage ± SE) | | | | | | | | | | Total aggregate |
|---|---|---|---|---|---|---|---|---|---|---|---|
| | Men | | | | | Women | | | | | |
| | Aggregate | <35 yrs | 35–59 yrs | >60 yrs | p-value* | Aggregate | <35 yrs | 35–59 yrs | >60 yrs | p-value* | |
| Daily smoking (n = 5593) | 516 (20.8±1.7) | 133 (17.2±2.6) | 281 (23.7±1.9) | 102 (29.9±4.1) | 0.01 | 372 (6.7±0.7) | 39 (1.4±0.3) | 238 (11.6±1.3) | 95 (22.7±2.8) | <0.001 | 888 (13.3±0.9) |
| Consumption of alcohol in last 12 months (n = 5593) | 876 (38.6±2.4) | 276 (35.7±3.2) | 489 (43.9±2.9) | 111 (33.7±4.1) | 0.03 | 469 (10.8±1.3) | 123 (6.7±1.2) | 278 (15.6±1.7) | 68 (17.6±3.7) | <0.001 | 1345 (23.8±1.3) |
| Insufficient physical activity (n = 5593) | 302 (16.0±1.8) | 90 (15.4±2.4) | 145 (14.5±1.8) | 67 (26.5±3.9) | 0.01 | 510 (14.6±1.5) | 206 (14.4±1.9) | 218 (13.8±1.8) | 86 (19.8±3.1) | 0.23 | 812 (15.2±1.3) |
| Insufficient fruits veg consumption (n = 5593) | 1639 (81.7±2.3) | 535 (84.1±2.6) | 841 (77.7±3.1) | 263 (84.3±3.0) | 0.05 | 3123 (85.1±2.1) | 1331 (83.8±2.8) | 1447 (86.4±1.8) | 345 (89.9±2.4) | 0.12 | 4762 (83.5±2.2) |
| Hypertension (n = 5593) | 775 (32.1±1.9) | 153 (23.7±2.6) | 452 (40±2.2) | 170 (50.1±3.9) | <0.001 | 867 (20.8±1.3) | 181 (9.9±1.2) | 506 (32.6±2.0) | 180 (46.0±3.3) | <0.001 | 1642 (26.1±1.1) |
| Diabetes (n = 5593) | 142 (6.0±0.9) | 24 (3.7±1.2) | 89 (8.8±1.2) | 29 (8.1±2.2) | 0.009 | 188 (4.9±0.6) | 40 (2.7±0.7) | 120 (8.0±1.1) | 28 (6.8±1.6) | <0.001 | 330 (5.4±0.6) |
| Overweight and obesity (n = 5519) | 530 (24.2±2.0) | 126 (19.1±2.5) | 334 (32.0±2.4) | 70 (21.7±4.2) | <0.001 | 1007 (25.2±1.6) | 357 (19.3±1.8) | 563 (35.5±2.0) | 87 (19.7±2.8) | <0.001 | 1537 (23.9±1.2) |
| Hyperlipidemia (n = 5593) | 191 (7.4±0.8) | 39 (4.2±0.8) | 123 (11.0±1.6) | 29 (11.4±2.6) | <0.001 | 499 (13.4±0.9) | 106 (7.5±1.1) | 288 (18.8±1.5) | 105 (31.6±3.3) | <0.001 | 690 (10.5±0.7) |

* p-value of the Chi-square test for differences between age groups

**Table 3. Effect of sociodemographic characteristics on behavioral risk factors—stratified by sex.** (Results from multiple mixed logistic regression models adjusting for all sociodempgraphic variables jointly).

| Sociodemographic variables | Daily Smoking | | | | Alcohol consumption in 12 months | | | |
|---|---|---|---|---|---|---|---|---|
| | Men | | Women | | Men | | Women | |
| | Odds Ratio (95% CI) | p-value* | Odds Ratio (95% CI) | p-value* | Odds Ratio (95% CI) | p-value* | Odds Ratio (95% CI) | p-value* |
| **Age Group** | | | | | | | | |
| Below 35 years | – | – | – | – | – | – | – | – |
| 35–59 years | 1.25 (0.95–1.64) | 0.11 | **3.80 (2.58–5.60)** | **<0.001** | 1.23 (0.95–1.57) | 0.10 | **2.28 (1.68–3.09)** | **<0.001** |
| 60 years and above | 1.34 (0.93–1.94) | 0.11 | **5.71 (3.63–9.00)** | **<0.001** | **0.56 (0.39–0.81)** | **0.002** | **2.26 (1.45–3.53)** | **<0.001** |
| **Ethnicity** | | | | | | | | |
| Dalits | – | – | – | – | – | – | – | – |
| Disadvantaged janjati | **0.52 (0.35–0.76)** | **<0.001** | **0.44 (0.29–0.66)** | **<0.001** | 0.87 (0.59–1.29) | 0.49 | 1.26 (0.83–1.91) | 0.83 |
| Religious minorities | **0.36 (0.15–0.83)** | **0.01** | **0.11 (0.02–0.54)** | **0.006** | **0.02 (0.05–0.13)** | **<0.001** | 0.06 (0.01–4.67) | 0.79 |
| Advantaged janajati | 0.66 (0.42–1.04) | 0.07 | **0.58 (0.37–0.92)** | **0.02** | 1.45 (0.93–2.12) | 0.10 | **2.27 (1.47–3.51)** | **<0.001** |
| Upper cast | 0.74 (0.51–1.08) | 0.11 | **0.63 (0.44–0.91)** | **0.01** | **0.63 (0.43–0.91)** | **0.01** | **0.15 (0.09–0.24)** | **<0.001** |
| **Educational attainment** | | | | | | | | |
| Primary incomplete | – | – | – | | – | – | – | – |
| Primary complete | **0.68 (0.50–0.93)** | **0.01** | **0.17 (0.10–0.30)** | **<0.001** | 0.84 (0.62–1.14) | 0.26 | **0.68 (0.47–0.99)** | **0.04** |
| Secondary | **0.50 (0.37–0.67)** | **<0.001** | **0.09 (0.04–0.19)** | **<0.001** | 0.79 (0.59–1.06) | 0.11 | **0.40 (0.27–0.60)** | **<0.001** |
| University | **0.08 (0.03–0.2)** | **<0.001** | 0.14 (0.62–3.29) | 0.30 | **0.46 (0.27–0.80)** | **0.005** | 0.67 (0.23–1.94) | 0.46 |
| **Location** | | | | | | | | |
| Rural location | – | – | – | – | – | – | – | – |
| Urban location | 0.77 (0.57–1.05) | 0.09 | 0.84 (0.60–1.17) | 0.30 | 0.82 (0.59–1.14) | 0.24 | 0.84 (0.53–1.35) | 0.48 |
| **Wealth Index** | | | | | | | | |
| Lower wealth index | – | – | – | – | – | – | – | – |
| Middle wealth index | **1.38 (1.02–1.87)** | **0.03** | **0.47 (0.33–0.68)** | **<0.001** | 1.23 (0.91–1.64) | 0.17 | 0.87 (0.62–1.23) | 0.40 |
| Upper wealth index | 1.02 (0.79–1.31) | 0.85 | **0.33 (0.24–0.46)** | **<0.001** | 1.01 (0.80–1.27) | 0.94 | **0.67 (0.51–0.89)** | **0.005** |
| Sociodemographic variables | Insufficient Fruits/veg consumption | | | | Insufficient Physical activity | | | |
| | Men | | Women | | Men | | Women | |
| | Odds Ratio (95% CI) | p-value* | Odds Ratio (95% CI) | p-value* | Odds Ratio (95% CI) | p-value* | Odds Ratio (95% CI) | p-value* |
| **Age Group** | | | | | | | | |
| Below 35 years | – | – | – | – | – | – | – | – |
| 35–59 years | 0.70 (0.48–1.02) | 0.06 | 0.92 (0.69–1.24) | 0.61 | 1.01 (0.72–1.42) | 0.91 | 0.95 (0.73–1.25) | 0.74 |
| 60 years and above | 0.75 (0.44–1.28) | 0.29 | 1.51 (0.89–2.53) | 0.12 | **2.04 (1.30–3.19)** | **<0.001** | **2.26 (1.54–3.31)** | **<0.001** |
| **Ethnicity** | | | | | | | | |
| Dalits | – | – | – | – | – | – | – | – |
| Disadvantaged janjati | 0.94 (0.46–1.92) | 0.87 | 1.20 (0.69–2.10) | 0.51 | 1.68 (0.95–2.95) | 0.06 | 1.12 (0.74–1.69) | 0.58 |
| Religious minorities | **0.09 (0.02–0.35)** | **<0.001** | 0.39 (0.15–1.03) | 0.06 | 2.11 (0.74–5.98) | 0.15 | 1.36 (0.59–3.12) | 0.46 |
| Advantaged janajati | 0.82 (0.40–1.69) | 0.59 | **0.47 (0.28–0.79)** | **0.004** | 1.75 (0.93–3.26) | 0.08 | 1.02 (0.63–1.64) | 0.94 |
| Upper cast | 0.65 (0.35–1.21) | 0.17 | **0.56 (0.36–0.88)** | **0.01** | 1.56 (0.90–2.67) | 0.10 | 1.00 (0.68–1.45) | 0.99 |
| **Educational attainment** | | | | | | | | |
| Primary incomplete | – | – | – | – | – | – | – | – |
| Primary complete | **0.42 (0.26–0.66)** | **<0.001** | 0.76 (0.53–1.10) | 0.15 | 1.27 (0.85–1.88) | 0.23 | 0.90 (0.65–1.26) | 0.56 |
| Secondary | **0.48 (0.31–0.75)** | **0.002** | **0.59 (0.41–0.85)** | **0.004** | 1.21 (0.82–1.78) | 0.33 | 0.89 (0.64–1.23) | 0.49 |
| University | **0.28 (0.13–0.59)** | **<0.001** | 0.85 (0.34–2.03) | 0.83 | 1.67 (0.85–3.28) | 0.13 | 0.82 (0.38–1.78) | 0.62 |
| **Location** | | | | | | | | |
| Rural location | – | – | – | – | – | – | – | – |
| Urban location | 1.28 (0.65–2.50) | 0.46 | 1.07 (0.56–2.03) | 0.96 | **1.76 (1.14–2.73)** | **0.01** | **2.08 (1.35–3.20)** | **<0.001** |

*(Continued)*

**Table 3.** (Continued)

| Wealth Index | | | | | | | | |
|---|---|---|---|---|---|---|---|---|
| Lower wealth index | – | – | – | – | – | – | – | – |
| Middle wealth index | 1.40 (0.89–2.19) | 0.14 | 1.11 (0.78–1.57) | 0.55 | 0.75 (0.50–1.13) | 0.21 | 1.08 (0.80–1.47) | 0.59 |
| Upper wealth index | 0.91 (0.65–1.28) | 0.59 | 1.27 (0.96–1.68) | 0.09 | 0.98 (0.72–1.34) | 0.93 | 0.96 (0.75–1.23) | 0.77 |

* p-value of the General Wald test

Munich Medical Research School, Ludwig Maximilians University (LMU) Munich, Germany (project no. 20–657). As the study represents a further analysis of the STEPS survey-2019 dataset, there was no need to obtain informed consent from study participants.

# Results

## Sociodemographic characteristics

The total sample population (n = 5593) had a mean age of 35.0 years (standard deviation SD = 14.4 years) with 64.3% of the sample being female (Table 1). More than one-third of the participants belonged to the upper caste and almost one-third were disadvantaged Janajati. Nearly half of the participants were illiterate or had not completed their primary education. More than half of the participants were engaged as homemakers and non-paid workers, and more than 60% were residing in an urban location **(Table 1)**.

Weighted proportions of sociodemographic characteristics illustrate that the sample population represents the Nepalese adult population very well when compared to the census report of 2011 [27]. Moreover, the weighted proportion of men and women was equally representative across the three age groups as illustrated in (**S2 File**).

## Prevalence of behavioral and biological risk factors

Daily smoking increased significantly with increasing age, among both sexes (Table 2). But the increase was far stronger in women, ranging from 1% prevalence for the youngest to 12% among the middle aged and 23% for the oldest age group. Among men, the prevalence of daily smoking increased from 17% among the youngest age group to 24% among the middle aged and 30% in the elderly. Similarly, alcohol consumption increased substantially with age among women from 7% prevalence in the youngest age group to 18% in the oldest age. In contrast, for men, prevalence of alcohol consumption was highest (44%) among 35–59 year-olds, followed by 38% among <35 year-olds and 34% among those aged 60 years and older. The highest prevalence of insufficient physical activity was found in the age group 60 years and above, higher among men than among women (27% vs. 20%), while there is not much difference in physical inactivity between sexes for other age groups. Insufficient fruit and vegetable consumption was very high for all age groups with about 5 out of 6 adults having two or less servings per day.

The prevalence of the two biological risk factors hypertension and hyperlipidemia showed significant differences between males and females. Hypertension was significantly more common in males ranging from 10% among those aged below 35 years, to 33% among those aged 35–59 years and 46% among those aged 60 years and above. Hyperlipidemia was more common in females, ranging from 8% among those aged below 35 years, to 19% among those aged 35–59 years and 32% among those aged 60 years and above. The prevalence of both increased substantially with age so that nearly half of the men of the oldest age group suffered from hypertension and nearly every third woman of this age had hyperlipidemia. Overweight and

**Table 4. Age effects on the relationship between four behavioral and four biological risk factors—stratified by sex.** (Results from multiple mixed logistic regression models with interaction of age and health behaviour).

| Predicting variables (Adjusted for sociodemographic variables) | Hypertension (Women) | | Hypertension (Men) | | Diabetes (Women) | | Diabetes (Men) | |
|---|---|---|---|---|---|---|---|---|
| | AOR (95% CI) | p-value# | AOR (95% CI) | p-value# | AOR (95% CI) | p-value# | AOR (95% CI) | p-value# |
| **Daily Smoking** | | | | | | | | |
| Daily smokers | 0.93 (0.35–2.43) | 0.88 | 1.00 (0.62–1.63) | 0.97 | 2.41 (0.28–2.04) | 0.88 | 1.17 (0.41–3.34) | 0.75 |
| Age (35–59 years)* | **2.95 (2.35–3.69)** | **<0.001** | **2.41 (1.79–3.26)** | **<0.001** | **4.15 (2.56–6.73)** | **<0.001** | **2.64 (1.41–4.97)** | **<0.001** |
| Age (60 years and above)* | **6.10 (4.38–8.50)** | **<0.001** | **4.21 (2.80–6.34)** | **<0.001** | **4.18 (2.10–8.32)** | **<0.001** | **3.56 (1.64–7.72)** | **0.001** |
| Daily smoking: age 35–59 years | 0.65 (0.23–1.79) | 0.4 | 0.88 (0.50–1.56) | 0.67 | 4.13 (0.04–3.79) | 0.43 | 1.00 (0.31–3.22) | 0.99 |
| Daily smoking: age 60+ years | 0.85 (0.29–2.50) | 0.7 | 0.88 (0.43–1.80) | 0.74 | 3.43 (0.03–3.65) | 0.37 | 0.75 (0.18–3.16) | 0.69 |
| **Consumption of Alcohol in 12 months** | | | | | | | | |
| Alcohol consumption in last 12 months | **1.82 (1.09–3.04)** | **<0.001** | 1.48 (0.98–2.23) | 0.06 | 2.88 (0.82–10.04) | 0.09 | 1.01 (0.40–2.53) | 0.97 |
| Age (35–59 years)* | **2.75 (2.17–3.48)** | **<0.001** | **2.33 (1.61–3.36)** | **<0.001** | **4.17 (2.45–7.10)** | **<0.001** | **3.14 (1.54–6.38)** | **0.001** |
| Age (60 years and above)* | **6.47 (4.66–9.00)** | **<0.001** | **4.97 (3.16–7.79)** | **<0.001** | **4.60 (2.25–9.42)** | **<0.001** | **4.01 (1.76–9.09)** | **<0.001** |
| Alcohol in 12 months: age 35–59 years | 0.86 (0.48–1.53) | 0.62 | 1.05 (0.64–1.70) | 0.83 | 0.59 (0.15–2.30) | 0.19 | 0.65 (0.23–1.84) | 0.42 |
| Alcohol in 12 months: age 60+ years | **0.46 (0.21–0.98)** | **0.04** | 0.69 (0.36–1.33) | 0.27 | 0.21 (0.03–1.41) | 0.11 | 0.46 (0.11–1.88) | 0.28 |
| **Insufficient Fruits and Veg Consumption** | | | | | | | | |
| Insufficient fruits and veg consumption | 0.99 (0.60–1.62) | 0.98 | 0.95 (0.55–1.63) | 0.86 | 0.90 (0.31–2.56) | 0.84 | 1.37 (0.40–4.75) | 0.61 |
| Age (35–59 years)* | **2.48 (1.42–4.35)** | **<0.001** | **2.69 (1.51–4.78)** | **<0.001** | **7.04 (2.41–20.53)** | **0.003** | **4.41 (1.28–15.12)** | **0.01** |
| Age (60 years and above)* | **5.97 (2.54–14.04)** | **<0.001** | **2.71 (1.22–6.00)** | **0.01** | 1.00 (0.10–9.15) | 0.99 | **5.28 (1.15–24.16)** | **0.03** |
| Insufficient fruits and veg: age 35–59 years | 1.15 (0.63–2.07) | 0.64 | 0.84 (0.45–1.56) | 0.58 | 0.50(0.16–1.57) | 0.16 | 0.53 (0.14–2.01) | 0.35 |
| Insufficient fruits and veg: age 60+ years | 9.96 (0.40–2.42) | 0.99 | 1.61 (0.69–3.79) | 0.26 | 4.25 (0.43–4.12) | 0.21 | 0.58 (0.11–3.01) | 0.51 |
| **Insufficient Physical Activity** | | | | | | | | |
| Insufficient Physical Activity | 0.89 (0.54–1.47) | 0.65 | 0.55 (0.29–1.03) | 0.06 | 1.72 (0.70–4.19) | 0.73 | 0.38 (0.08–1.86) | 0.23 |
| Age (35–59 years)* | **2.73 (2.16–3.44)** | **<0.001** | **2.24 (1.69–2.96)** | **<0.001** | **4.15 (2.47–6.97)** | **<0.001** | **2.13 (1.20–3.79)** | **0.009** |
| Age (60 years and above)* | **5.60 (4.02–7.81)** | **<0.001** | **3.93 (2.68–5.77)** | **<0.001** | **4.73 (2.35–9.51)** | **<0.001** | **3.07 (1.49–6.31)** | **0.002** |
| Insufficient Physical activity: age 35–59 years | 1.22 (0.68–2.20) | 0.49 | 1.46 (0.70–3.04) | 0.30 | 0.82 (0.29–2.28) | 0.70 | 4.10 (0.76–22.17) | 0.10 |
| Insufficient Physical activity: age 60+ years | 1.33 (0.65–2.72) | 0.42 | 1.50 (0.64–3.53) | 0.34 | 0.30 (0.07–1.28) | 0.10 | 1.95 (0.28–13.20) | 0.49 |
| **Predicting Variables** (adjusted for sociodemographic variables) | Overweight/obesity (Women) | | Overweight/obesity (Men) | | Hyperlipidemia (Women) | | Hyperlipidemia (Men) | |
| | AOR (95% CI) | p-value# | AOR (95% CI) | p-value# | AOR (95% CI) | p-value# | AOR (95% CI) | p-value# |
| **Daily Smoking** | | | | | | | | |

(Continued)

**Table 4.** (Continued)

| | | | | | | | | |
|---|---|---|---|---|---|---|---|---|
| Daily smokers | 0.74 (0.28–1.91) | 0.54 | 1.29 (0.75–2.19) | 0.34 | 1.58 (0.50–4.92) | 0.42 | 1.11 (0.46–2.68) | 0.80 |
| Age (35–59 years)* | **1.63 (1.33–2.00)** | **<0.001** | **2.18 (1.56–3.05)** | **<0.001** | **2.68 (2.01–3.58)** | **<0.001** | **2.01 (1.21–3.36)** | **0.007** |
| Age (60 years and above)* | 1.17 (0.82–1.66) | 0.37 | 1.49 (0.93–2.38) | 0.09 | **4.56 (3.05–6.81)** | **<0.001** | 1.65 (0.82–3.36) | 0.15 |
| Daily smoking: age 35–59 years | 0.58 (0.21–1.60) | 0.29 | 0.67 (0.36–1.26) | 0.22 | 0.53 (0.16–1.77) | 0.30 | 0.73 (0.27–2.00) | 0.55 |
| Daily smoking: age >= 60 years | 0.35 (0.10–1.19) | 0.09 | 0.68 (0.29–1.60) | 0.38 | 0.83 (0.23–2.93) | 0.77 | 0.77 (0.21–2.76) | 0.68 |
| **Consumption of Alcohol in 12 months** | | | | | | | | |
| Alcohol consumption in last 12 months | **2.36 (1.50–3.71)** | **<0.001** | 1.21 (0.76–1.92) | 0.40 | **2.13 (1.11–4.09)** | **0.02** | 1.85 (0.88–3.88) | 0.10 |
| Age (35–59 years)* | **1.63 (1.32–2.01)** | **<0.001** | **2.17 (1.45–3.25)** | **<0.001** | **2.72 (2.02–3.65)** | **<0.001** | **2.39 (1.23–4.64)** | **0.009** |
| Age (60 years and above)* | 1.13 (0.79–1.61) | 0.47 | 1.28 (0.75–2.16) | 0.35 | **5.49 (3.68–8.20)** | **<0.001** | **2.42 (1.10–5.30)** | **0.02** |
| Alcohol in 12 months: age 35–59 years | **0.48 (0.28–0.82)** | **0.04** | 0.84 (0.49–1.45) | 0.54 | 0.59 (0.28–1.22) | 0.15 | 0.63 (0.27–1.47) | 0.29 |
| Alcohol in 12 months: age >= 60 years | **0.26 (0.11–0.61)** | **0.001** | 1.29 (0.60–2.76) | 0.50 | **0.37 (0.15–0.94)** | **0.03** | 0.34 (0.10–1.15) | 0.08 |
| **Insufficient Fruits and Veg Consumption** | | | | | | | | |
| Insufficient fruits and veg consumption | 0.69 (0.47–1.02) | 0.06 | 1.13 (0.61–2.09) | 0.68 | 1.34 (0.68–2.63) | 0.39 | 0.60 (0.25–1.44) | 0.25 |
| Age (35–59 years)* | 1.00 (0.62–1.62) | 0.11 | **2.51 (1.30–4.83)** | **0.005** | **4.36 (2.10–9.08)** | **<0.001** | 1.66 (0.68–4.02) | 0.25 |
| Age (60 years and above)* | 0.97 (0.40–2.36) | 0.95 | 2.03 (0.80–5.13) | 0.13 | **3.42 (1.08–10.84)** | **0.03** | 1.11 (0.29–4.25) | 0.86 |
| Insufficient fruits and veg: age 35–59 years | 1.61 (0.97–2.67) | 0.06 | 0.74 (0.36–1.50) | 0.41 | 0.54(0.25–1.18) | 0.12 | 1.13 (0.42–3.03) | 0.79 |
| Insufficient fruits and veg: age >= 60 years | 0.98 (0.39–2.49) | 0.97 | 0.60 (0.22–1.64) | 0.32 | 1.42(0.43–4.66) | 0.56 | 1.47 (0.34–6.29) | 0.59 |
| **Insufficient Physical Activity** | | | | | | | | |
| Insufficient Physical Activity | 1.04 (0.70–1.54) | 0.81 | 1.36 (0.76–2.46) | 0.29 | 0.75(0.39–1.44) | 0.39 | 0.42 (0.11–1.51) | 0.18 |
| Age (35–59 years)* | **1.54 (1.25–1.90)** | **<0.001** | **2.04 (1.48–2.80)** | **<0.001** | **2.38 (1.77–3.18)** | **<0.001** | **1.79 (1.11–2.88)** | **0.01** |
| Age (60 years and above)* | 0.91 (0.63–1.31) | 0.61 | 1.42 (0.91–2.32) | 0.11 | **4.06 (2.70–6.09)** | **<0.001** | 1.33 (0.68–2.58) | 0.39 |
| Insufficient Physical activity: age 35–59 years | 0.86 (0.52–1.44) | 0.58 | 0.82 (0.40–1.67) | 0.58 | 1.86 (0.88–3.92) | 0.09 | 1.61 (0.39–6.68) | 0.50 |
| Insufficient Physical activity: age >= 60 years | 1.14 (0.55–2.35) | 0.71 | 0.70 (0.28–1.78) | 0.46 | 2.36 (0.99–5.6) | 0.05 | 2.80 (0.57–13.78) | 0.20 |

AOR: Adjusted Odds Ratio (adjusted for sociodemographic variables: ethnicity, occupation, education, wealth index, location)

AOR: Adjusted Odds Ratio from multiple mixed logistic regression adjusted for sociodemographic variables: ethnicity, occupation, education, wealth index, location

* Reference age group: 15–35 years

# p-value of the General Wald test

obesity was about equally prevalent in men and women, with the prevalence being highest in the middle-aged population (35–59 years), where one out of three adults was affected. Also, the prevalence of hyperglycemia was highest among 35–59 years old (9% in men and 8% in women), followed by the oldest age group (8% in men and 7% in women) (**Table 2**).

While the prevalence of biological risk factors in Nepal is high, levels of diagnosis and treatment remain very low. Only 13% of the respondents with hypertension reported taking antihypertensive medicines, and only one out of 10 diabetes patients reported taking hypoglycemic medicines. For hyperlipidemia the treatment rate was as low as 2%. The treatment rate was lower among women compared to men, and lowest in women and men aged 60 years and older.

### Age and sociodemographic effects on behavioral risk factors

A multiple regression model including all sociodemographic variables jointly showcased other relevant characteristics associated with health behaviors and biological risk factors.

*Daily smoking* significantly increased with age among women, but not among men. In addition, daily smoking significantly differed among ethnic groups, being highest among Dalits. While a higher educational attainment reduced the prevalence of smoking in both sexes, higher wealth only reduced smoking among women; in contrast the chances of smoking were 38% higher among men of a middle wealth index compared to men of a low wealth index.

*Alcohol consumption* was significantly more prevalent among women of higher age groups, yet it was significantly lower among men aged 60 years and above. Apart from this, alcohol consumption showed a similar but less pronounced picture as described for tobacco use above. The ethnic group with the highest prevalence of alcohol consumption was the advantaged Janajati.

*Insufficient fruit and vegetable consumption* significantly differed between ethnic groups. For men it was lowest in religious minorities, while for women it was lowest among the advantaged Janajati and upper caste. Higher educational attainment reduced the prevalence of insufficient fruit and vegetable consumption among men, less so among women.

*Insufficient physical activity* was highest in the oldest age group for both, men and women, and among those living in an urban location (**Table 3**).

### Effects of health behaviors on biological risk factors and their associations with age

The risk of **hypertension** was found to be associated with alcohol consumption among men after being adjusted for sociodemographic factors. However, this effect was evened for the oldest age group. Besides that, age affected the prevalence of hypertension as described above.

After adjustment for sociodemographic variables, the risk of **diabetes** was not significantly associated with any of the behavioral risk factors but increased with age.

The prevalence of **overweight and obesity** was found to be significantly associated with alcohol consumption for young women. But this effect was leveled out again by the age interaction for the two older age groups. The risk of overweight and obesity was highest among 35–59 years old.

Also, the prevalence of **hyperlipidemia** was found to be significantly associated with alcohol consumption for young women. The risk of suffering from hyperlipidemia increased with age, and this age effect was stronger among women than among men. The effect of daily smoking on hyperlipidemia was not statistically significant, which could be due to the masking effect of age on hyperlipidemia. The interaction of age and alcohol consumption for the two older age groups was found to reduce this increased risk of hyperlipidemia. Also, the effect of physical activity on hyperlipidemia among women significantly differ between the youngest age group (below 35 years) and 60 years and above aged women (**Table 4**).

## Discussion

### Key findings

Existing literature suggests a higher prevalence of NCDs and their risk factors among the older population but the modifying effect of age on the associations between behavioral and biological risk factors for NCDs has not received much attention to date. This study, to our knowledge the first of its kind in LMICs like Nepal, shows that:

The prevalence of selected behavioral risk factors for NCDs, notably smoking, alcohol consumption and insufficient physical activity, increases with age. The prevalence of the biological risk factors hypertension and hyperlipidemia significantly increases with age. Among the elderly, i.e. those aged 60 years and above, hypertension affects every third man and more than one in five women; in addition, more than a third of elderly women suffer from hyperlipidemia. Overweight and obesity as well as diabetes are significantly more prevalent among 35–59 year old Nepalese adults. In this middle-aged population, every third person is overweight or obese and 8% and more suffer from diabetes. This finding might point towards a generation effect.

In our study, the most important factor influencing the prevalence of behavioral and biological risk factors for NCDs, is age. In contrast, age interactions in the association between behavioral risk factors and biological risk factors were surprisingly weak, with only alcohol consumption in women showing a significant impact on the biological risk factors hypertension, overweight and obesity and hyperlipidemia. Levels of diagnosis of and treatment for these biological risks are, however, very low in Nepal.

### Results in context with existing findings

In line with our findings previous studies in LMICs have indicated that older populations of lower socioeconomic status are more likely to use tobacco, alcohol and eat unhealthy diets [28, 29]. A European study showed overweight/obesity and physical inactivity to be the most prevalent NCD risk factors [30], while our study revealed overweight/obesity and hyperlipidemia for women and alcohol consumption and hypertension for men as the most common NCD risk factors in Nepal. The findings of previous studies among adults from eleven European countries [31] are similar to our study, showing that behavioral risk factors are more prevalent among men while biological risks factors, specifically overweight/obesity and hyperlipidemia, tend to be more common among women.

Previous studies revealed that hyperlipidemia is more common among smokers [32]. A randomized clinical trial in the USA suggested that current smoking is associated with raised total cholesterol and triglycerides in the blood [33] while our study suggests that risk of hyperlipidemia increases with age but not with smoking.

Hypertension was observed to only be determined by age, and no relationships with other risk factors were observed except for alcohol consumption in the last 12 months among women. Similar findings were reported in an Indonesian study from 2018, which suggested that there is no relationship between smoking and hypertension [34]. This, however, is likely due to limitations of the cross-sectional study design, as longitudinal studies show a clear relationship between smoking and hypertension [35].

The effect of alcohol consumption on blood pressure may vary between males and females, indicating that low to moderate alcohol consumption may have a protective effect in females [36]. In contrast, our study examined the effect of alcohol consumption in three different age groups and suggests that the risk of hypertension is significantly increased by alcohol consumption in women. This association between alcohol consumption and hypertension,

however is not observed for the older age group, where the effect is leveled out by the interaction effect with age.

A meta-analysis investigating the effects of alcohol consumption on overweight and obesity showed varying results. A cohort study showed that there was no significant association between drinking alcohol and overweight and obesity but in cross-sectional studies, alcohol intake was found to be associated with overweight [37]. Our study shows a significant association between alcohol consumption and overweight and obesity in young women. This is, however, not observed in the older age groups or in men.

A study in China suggested that the prevalence of hyperlipidemia was positively correlated with alcohol consumption [38], which is supported by the findings of our study showing a strong association between alcohol consumption and hyperlipidemia in young women.

## Strengths and limitations

A major strength of this study is that it is based on a large, well designed and nationally representative survey using the WHO standardized STEPS survey tool. This survey offers an elaborate survey design that ensures representativeness of the data with negligible missing values, and guarantees a high degree of quality control during data collection and data entry. Another point that contributes to the robustness of the results is the underlying, well-defined conceptual framework that guided our data analysis. All biological measures were taken by trained health care workers, thus ensuring higher quality than self-reported data would.

However, a limitation of the data set with regards to the impact of age is the cross-sectional nature of the data, which makes it difficult to disentangle effects between different generations from differences between different age groups. The statistical analysis did not use age as a continuous variable but as a categorical variable comprising three broad age groups. Among women, this may create a problem, as it may yield confounded effects of menopausal symptoms and hormonal changes among middle-aged women. To conclude on a causal effect of age longitudinal data would be needed. Also, the data is limited to the population up to age 69, thus ignoring the oldest within the Nepalese population. Additional weaknesses relate to the measurement of the behavioral variables, which are all self-reported. Tobacco consumption records only if any tobacco products are currently used but does not differentiate between smoking and use of smokeless tobacco products, nor does it measure the amount used. Given the large number of variables included in our analysis, we had to limit ourselves to one measure of tobacco and chose the one that would likely show the greatest effect. Alcohol consumption only documents any alcohol consumed within the last twelve months and thus might also be a proxy for general lifestyle and health behavior considering the multi-ethnical population of Nepal. The measure of fruit and vegetable consumption is unsatisfactory, as it does not allow a small degree in dietary variation to be captured. Consumption of fruit and vegetables is known to be very low in Nepal, and given the limited variation within the population, no significant effects are to be expected.

## Conclusion

In Nepal, based on the findings of this study, age and sex exert a strong effect on the prevalence of behavioral and biological risk factors for NCDs and, ultimately, on the prevalence of NCDs. The WHO "25 by 25" goal requires a 25% reduction in premature NCD mortality by 2025. Our study suggests that one of the most important targets for Nepal is detection and effective treatment of the biological risk factors; hypertension, hyperlipidemia and diabetes, among younger age groups–notably among those aged 35–59 years, before they manifest as NCDs. The health system will need to react now and in planning for the future by paying greater

attention to train health workers at all levels, and by appropriately equipping health facilities for the effective management of risk factors. In addition, all NCD risk factors will need to be tackled early in life among those aged less than 35 years and throughout life through both education and creating enabling physical, social and policy environments that allow people to pursue a healthy lifestyle. Preventing unhealthy behaviors among younger age groups will reduce the risk of biological risk factors and ultimately of NCDs in older age. Notably, ensuring a healthy diet for the younger generation will be an important means to prevent overweight and obesity in generations to come.

## Supporting information

**S1 File. Definition and measurement of variables under study.**
(DOCX)

**S2 File. Weighted proportion of sociodemographic variables indicating the true representation of the study population.**
(DOCX)

## Acknowledgments

We would like to thank Mr. Bikram Adhikari from Kathmandu University, Dhulikhel Hospital for his guidance while analyzing the data by using R-studio. We are grateful to Ms. Marie-Elssa Morency, Regional Cancer Care Program Coordinator, British Columbia, Canada for language editing the manuscript.

## Author Contributions

**Conceptualization:** Bhim Prasad Sapkota.

**Data curation:** Bhim Prasad Sapkota.

**Formal analysis:** Bhim Prasad Sapkota, Ursula Berger.

**Methodology:** Bhim Prasad Sapkota.

**Project administration:** Bhim Prasad Sapkota.

**Supervision:** Kedar Prasad Baral, Eva A. Rehfuess, Klaus G. Parhofer, Ursula Berger.

**Writing – original draft:** Bhim Prasad Sapkota.

**Writing – review & editing:** Bhim Prasad Sapkota, Kedar Prasad Baral, Eva A. Rehfuess, Klaus G. Parhofer, Ursula Berger.

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
