## [Decision Letter · Decision Letter 0]

9 Mar 2023

PONE-D-23-01133

Effects of age on non-communicable disease risk factors among Nepalese adults

PLOS ONE

Dear Dr. Sapkota,

Thank you for submitting your manuscript to PLOS ONE. After careful consideration, we feel that it has merit but does not fully meet PLOS ONE’s publication criteria as it currently stands. Therefore, we invite you to submit a revised version of the manuscript that addresses the points raised during the review process.

Please submit your revised manuscript by Apr 23 2023 11:59PM.  If you will need more time than this to complete your revisions, please reply to this message or contact the journal office at plosone@plos.org. Please include the following items when submitting your revised manuscript:

While addressing comments from reviewers, also ensure that your manuscript meets PLOS ONE's style requirements, including those for file naming. The PLOS ONE style templates can be found at

We look forward to receiving your revised manuscript.

Kind regards,

Achyut Raj Pandey,

Academic Editor

PLOS ONE

5. Please include your tables as part of your main manuscript and remove the individual files. Please note that supplementary tables (should remain/ be uploaded) as separate "supporting information" files

Reviewers' comments:

Reviewer's Responses to Questions

**Comments to the Author**

1. Is the manuscript technically sound, and do the data support the conclusions?

Reviewer #1: Yes

Reviewer #2: Yes

Reviewer #3: Yes

2. Has the statistical analysis been performed appropriately and rigorously?

Reviewer #1: I Don't Know

Reviewer #2: I Don't Know

Reviewer #3: Yes

3. Have the authors made all data underlying the findings in their manuscript fully available?

Reviewer #1: Yes

Reviewer #2: Yes

Reviewer #3: No

4. Is the manuscript presented in an intelligible fashion and written in standard English?

Reviewer #1: Yes

Reviewer #2: Yes

Reviewer #3: Yes

5. Review Comments to the Author

Reviewer #1: The authors have done an interesting study to see how the risk factors of NCDs vary with age which will add some knowledge in the existing information of the NCDs in Nepal. The authors can revise the following information in their submission.

1. In the supporting information S1, the educational attainment of respondents has been classified as : Illiterate, Primary level, Secondary Level and University level which does not match with that of the names used in the table under supporting information S2 that has “below primary”, “primary completed”, “secondary level” and “University Education”. Please make consistent where necessary

2. Does “others” under the heading “occupation” refer to “Non paid workers and retired” as stated in S1? Please look for similar inconsistencies.

3. Under ethical approval heading in the main document, please write full form of NHRC and LMU when they are used for the first time as an acronym in the main document.

4. Fix some typo; change cast to caste in S1.

5. The age has been broadly categorized into three groups i.e. <35 years, 35-59 years and >=60 years which seems to be a major limitation of the study as this broad categorization can not give a reliable estimate of how age modifies the effect of health behavior on biological risk factors. This creates a problem particularly for analysis of women’s data because of the confounded effect of menopausal symptoms and hormonal changes during the 40’s among women. The authors might consider adding a limitation on why they could not measure age on a continuous scale to measure its effect on non-communicable disease disk factors.

6. I could not see tables 2,3,4. Please revisit.

Reviewer #2: Abstract word count is exceeding the limit as per journal guideline.

Please remove the bullet points from the discussion. Paragraphs would be better.

Was there any effect of age on smoking?

Plausible reason for hyperlipidemia increasing with age but not with smoking in this study needs to be explained.

Reviewer #3: The authors Sapkota et al have attempted to portray the picture of NCD risk factors in their linkage with age among Nepalese adults which has come with a good effort and a good analysis and write up. To make this article a good value addition to the science, it would be great if the following comments be addressed with some additional rigor. Please find specific comments below.

Introduction:

The references (esp. #1-3) used in the introduction are older which could be replaced with newer latest evidence and edition of the same or similar reports. E.g., you could use the progress monitor report available for 2022 - Noncommunicable Diseases Progress Monitor 2022 (who.int)); furthermore, it’s better to use the global status report rather than citing a big statement about the situation of NCDs from the global action plan for prevention and control of NCDs and that is older as well; and the latest one of the global status report could also be used.

Check if the ref #4 is listed correctly in the references list. Please check other references as well and make sure they are complete. For e.g., # 9 looks bit incomplete. Other references also seem to have been written incorrectly (e.g. 12, 13); Nepal STEPS 2019 report could be referenced more appropriately (also it has been listed in the ref list twice). and so on… hence please check all the references thoroughly.

In the beginning of second para the information cited with ref #9 looks to be old (2009 report?). There are several latest evidence available for this as well.

For reporting the latest life expectancy of Nepal you could either use world bank data or GBD 2019 data as reference.

Methods:

In the behavioral risk factors, why only daily smoking and no other forms or tobacco use was considered including non-daily smoking?

Biological risk factors – better to specify cholesterol as total cholesterol.

Results:

In the section prevalence of behavioral and biological risk factors the second line talks about tobacco among women in different age groups but compares at the end with single group of men which looks little irrelevant comparison, please clarify/amend.

In the same section, in second para you suddenly start using hypertension and hyperlipidemia while your variables are raised BP and raised total cholesterol. Suggest to stick to original variables, and also hyperlipidemia may not be the right choice of word since we are just talking about total cholesterol and not other cholesterols. And follow the use of same name for risk factors in rest of the manuscript.

In addition, while describing the results by sex I would always add age disaggregated description (which you have done in some places but not in all, please follow this to justify your title and main objective of the study) since that is your main variable of interest. The difference of prevalence of risk factors in general is already described by the report and manuscript of the STEPS survey. Please follow the same in the third para of this section – on the treatment seeking behavior part.

In the last two sections of results you have attempted to describe the effects of age, which is commendable and is the beauty of this manuscript. However, this could be further beautified by being more consistent and coherent in the description. Try to ensure you describe the relationship or effects of other variables on your dependent variables and then describe what and how age has affected that. You have done in some part, but readers will love it if you do that consistently and coherently. Try to read it once from this angle, and this will be done easily.

Strengths and limitations

The last line about biological measures look to be written as limitation, but I believe you meant that is the strength, so revise that to reflect that you wanted to say that it was one of the strengths of the study.

Conclusion:

Your conclusion is more generic towards health systems improvement in most of the statements except few where you have touched the age part. Readers would love to see some more specific conclusion/recommendation based on your specific significant results of your analysis of age effect. That would make the conclusion part even better and precise.

Abstract:

I recommend revising the conclusion part in the summary as well after you are done revising your paper as above.

6. PLOS authors have the option to publish the peer review history of their article (what does this mean?). If published, this will include your full peer review and any attached files.

**Do you want your identity to be public for this peer review?** For information about this choice, including consent withdrawal, please see our Privacy Policy.

Reviewer #1: No

Reviewer #2: **Yes: **Pranil Man Singh Pradhan

Reviewer #3: **Yes: **Krishna Aryal

While revising your submission, please upload your figure files (if applicable) to the Preflight Analysis and Conversion Engine (PACE) digital diagnostic tool, https://pacev2.apexcovantage.com/. PACE helps ensure that figures meet PLOS requirements. To use PACE, you must first register as a user. Registration is free. Then, login and navigate to the UPLOAD tab, where you will find detailed instructions on how to use the tool. If you encounter any issues or have any questions when using PACE, please email PLOS at figures@plos.org. Please note that Supporting Information files do not need this step.

---

## [Author Response · Author response to Decision Letter 0]

24 Apr 2023

The comments from reviewers have been incorporated and submitted in the tabular form.

---

## [Decision Letter · Decision Letter 1]

12 May 2023

Effects of age on non-communicable disease risk factors among Nepalese adults

PONE-D-23-01133R1

Dear Dr. Sapkota,

We’re pleased to inform you that your manuscript has been judged scientifically suitable for publication and will be formally accepted for publication once it meets all outstanding technical requirements.

Kind regards,

Achyut Raj Pandey, MPH

Academic Editor

PLOS ONE

Additional Editor Comments (optional):

Reviewers' comments:

Reviewer's Responses to Questions

**Comments to the Author**

1. If the authors have adequately addressed your comments raised in a previous round of review and you feel that this manuscript is now acceptable for publication, you may indicate that here to bypass the “Comments to the Author” section, enter your conflict of interest statement in the “Confidential to Editor” section, and submit your "Accept" recommendation.

Reviewer #1: All comments have been addressed

Reviewer #2: All comments have been addressed

Reviewer #3: All comments have been addressed

2. Is the manuscript technically sound, and do the data support the conclusions?

Reviewer #1: Yes

Reviewer #2: Yes

Reviewer #3: Yes

3. Has the statistical analysis been performed appropriately and rigorously? 

Reviewer #1: Yes

Reviewer #2: Yes

Reviewer #3: Yes

4. Have the authors made all data underlying the findings in their manuscript fully available?

Reviewer #1: Yes

Reviewer #2: No

Reviewer #3: Yes

5. Is the manuscript presented in an intelligible fashion and written in standard English?

Reviewer #1: Yes

Reviewer #2: Yes

Reviewer #3: Yes

6. Review Comments to the Author

Reviewer #1: Thank you for duly addressing the comments. Although it definitely has some limitations, the research tries to add some knowledge in the existing literature related to non-communicable diseases in Nepal.

Reviewer #2: (No Response)

Reviewer #3: Thank you for addressing the comments. The manuscript reads well now, and the revision is done satiefactorily.

7. PLOS authors have the option to publish the peer review history of their article (what does this mean?). If published, this will include your full peer review and any attached files.

Reviewer #1: No

Reviewer #2: No

Reviewer #3: **Yes: **Krishna Kumar Aryal

---

## [Editor Report · Acceptance letter]

22 May 2023

PONE-D-23-01133R1 

Effects of age on non-communicable disease risk factors among Nepalese adults 

Dear Dr. Sapkota:

I'm pleased to inform you that your manuscript has been deemed suitable for publication in PLOS ONE. Congratulations! Your manuscript is now with our production department. 

Kind regards, 

on behalf of

Mr. Achyut Raj Pandey 

Academic Editor

PLOS ONE